# Physical Impact of a Typical Training Session with Different Volumes on the Day Preceding a Match in Academy Soccer Players

**DOI:** 10.3390/ijerph192113828

**Published:** 2022-10-24

**Authors:** Tom Douchet, Christos Paizis, Nicolas Babault

**Affiliations:** 1Center for Performance Expertise, Sport Science Faculty, CAPS, U1093 INSERM, University of Bourgogne-Franche-Comté, 3 Allée des Stades Universitaires, BP 27877, CEDEX, 21078 Dijon, France; 2Dijon Football Côte d’Or (DFCO), 17 rue du Stade, 21000 Dijon, France

**Keywords:** tapering, workload, performance, micro-cycle, youth

## Abstract

French academies almost all implement reactivity drills and small-sided games (SSG) the day preceding a match (MD-1). The present study aimed to determine the physical impact of different training durations on MD-1 on the subsequent matchday performance (MD). Eleven elite U19 academy soccer players conducted three typical training sessions lasting 45 min (TS45), 60 min (TS60) and 75 min (TS75) on MD-1. During TS60, warm-up, reactivity and SSG were 10, 15 and 24 min, respectively, plus coaches’ feedback or water breaks. Durations of all training components decreased by 25% for TS45 and increased by 25% for TS75. Tests were conducted on MD-4 (CONTROL) and MD before the match (TEST). Tests consisted of a counter movement jump (CMJ), 20 m sprint, Illinois agility test (IAT) and Hooper questionnaire. CONTROL values were similar over the three experimental conditions. TEST on MD revealed greater CMJ for TS45 (42.7 ± 5.1 cm) compared to TS60 (40.5 ± 5.5 cm, *p* = 0.032) and TS75 (40.9 ± 5.7 cm, *p* = 0.037). 20 m time was lower for TS45 (3.07 ± 0.10 s, *p* < 0.001) and TS60 (3.13 ± 0.10 s, *p* = 0.017) compared to TS75 (3.20 ± 0.10 s). IAT time was lower on TS45 (14.82 ± 0.49 s) compared to TS60 (15.43 ± 0.29 s, *p* < 0.001) and TS75 (15.24 ± 0.33 s, *p* = 0.006). Furthermore, the Hooper index was lower at TEST for TS45 (7.64 ± 1.50) compared to TS60 (11.00 ± 3.49, *p* = 0.016) and TS75 (9.73 ± 2.41, *p* = 0.016), indicating a better readiness level. We concluded that, as training session duration increases, performance decreases on MD. A 45 min training session including reactivity and SSG exercises is therefore recommended on MD-1.

## 1. Introduction

Periodization is one of the main questions facing training staff working with elite athletes [1]. To date, there are no guidelines on how to manage the weekly training load in soccer. Most of the studies focusing on weekly periodization have depicted practices without objectifying them [2,3,4]. A common trend across teams is generally reported, with the greatest weekly training loads usually occurring mid-way between two consecutive matches [5]. Nevertheless, the magnitude of the training load varies between teams [6] and is partly attributed to the coach’s background and national habits [7].

On the day preceding the match (MD-1), a workload decrease is commonly observed in professionals [2,6,8] and youth [3,4]. Indeed, players must be in optimal readiness to play and withstand a heavy competitive demand [9]. Therefore, this training load decrease can be considered an intra-week tapering method to favor recovery from mid-week sessions and competitive performance [10].

In a recent survey in elite French soccer academies [11], most practitioners reported using a short training session for tapering on MD-1. This session was mainly composed of reactivity drills and small-sided games (SSG) for 60 min [11]. Reactivity drills have already been reported in previous studies without justification [2,12,13]. A recent study indicated that this session was concluded by set pieces due to their tactical importance and low energy requirement [2]. The use of SSG so close to competition can be surprising. Indeed, SSG could potentially lead to performance decrements (here defined as fatigue [14] through elevated plasma creatine kinase [15] and blood lactate [16]), as a consequence of the numerous accelerations, decelerations and high-speed running [17]. It was shown that implementing larger pitch sizes elicited greater hamstring fatigue while smaller pitch sizes could lower the training load [18]. SSG have been extensively studied and studies have shown that training load could also be adjusted by manipulating multiple variables (e.g., volume expressed as work to rest ratio) [17]. For instance, the continuous format of SSG would lead to a greater physical load compared to the same duration in intermittent format [19].

To our knowledge, this popular MD-1 training session, usually conducted across elite French soccer academies, has never been tested and is more likely to be a practical, habit-based session. Therefore, in this study, we aimed to explore the effects of such a training session using various training durations (i.e., training volume). We hypothesized that the current duration of the MD-1 session would decrease the physical and psychological characteristics of the players on the MD and decreasing this duration might increase performance. The results from the present study would have a large impact for practitioners regarding training periodization to promote soccer performance and for injury prevention.

## 2. Materials and Methods

### 2.1. Participants

Twenty elite academy soccer players (age = 18.2 ± 0.4 years, height = 177.5 ± 1.6 cm, body mass = 72.1 ± 2.3 kg, body fat = 9.7 ± 1.2%) of the U19 team of the elite French soccer club of Dijon Football Côte d’Or (DFCO) were recruited for this study. The sample size was calculated a priori using G*Power (version 3.1.9.6, free software available at https://www.psychologie.hhu.de/arbeitsgruppen/allgemeine-psychologie-und-arbeitspsychologie/gpower.html, accessed on 8 August 2022) with the following criteria: effect size of 0.3, power of 0.8, probability error of 0.05. A sample size of 20 players was indicated. Goalkeepers were excluded from this study due to the different nature of their activity. Therefore, 4 central defenders, 6 full backs, 4 central midfielders, 3 wide midfielders and 3 forwards were included. Only players completing every session during the protocol were included in the study. Players were instructed to maintain their regular nutritional status. All the players were notified of the research protocol, benefits and risks before providing written informed consent in accordance with the declaration of Helsinki. Approval of the study was obtained by the local ethic committee (CERSTAPS, a national ethics research committee in sports sciences: IRB00012476-2021-22-04-105).

### 2.2. General Design

This study was conducted during the 2020–2021 in-season for seven consecutive weeks, before the winter break, between the 14th and 19th week of the season. All sessions took place on an artificial field. During the first week, participants conducted a familiarization session (Figure 1). During this session, participants were familiarized with all physical tests, Hooper Questionnaire and training contents. Then, three experimental weeks were conducted and were interspersed with three standardized training weeks. To obtain a similar fitness level, the standardized training weeks were strictly identical in content, duration and intensity. Therefore, training sessions from MD+2 to MD-2 were identical. External and internal load were monitored during these sessions to ensure that training load was similar (not reported here). Briefly, MD+2 consisted in a recovery-oriented session implementing aerobic technical exercises for 60 min. MD-4 consisted in a strength-oriented session with small-sided games for 90 min. MD-3 was an aerobic-oriented session with medium- to large-sided games for 90 min. MD-2 was a speed-oriented session using large-sided games for 75 min. Similarly, the content, duration and intensity of all training sessions were identical during the experimental weeks (Figure 1). The only difference between experimental weeks was on MD-1. During this training session (MD-1), training content and intensity were similar, but duration was either 45 min (TS45), 60 min (TS60) or 75 min (TS75). Training was composed of a standardized warm-up, reactivity work and SSG. Reactivity work consisted of a 5-0-5 test [20] with an auditive start signal. Runs were interspersed by 2 min of recovery. SSG was 5 against 5 plus goalkeepers on a 30 × 40 m wide field (representing 120 m^2^ for each player). Sequences were 2 min work and 2 min passive rest. This method of SSG was chosen in accordance with responses given by practitioners who answered the survey demonstrating that French academies typically use these drills on MD-1 [11]. During TS60, warm-up, reactivity and SSG were 10 min, 15 min and 24 min, respectively. The remaining time was used for water breaks or coaches’ feedback. During TS45, training duration was decreased by 25%: 8 min of warm-up, 12 min reactivity work and 16 min SSG. During TS75, training duration was increased by 25%: 12 min of warm-up, 20 min reactivity work and 32 min SSG. Players were blinded from training duration of the training session they were performing and from the different duration tested throughout the study to prevent any pacing strategy.

Players were tested during two separate days of the experimental weeks. Tests on MD-4 served as CONTROL to verify that players were in a similar fitness state. Tests on MD (here called TEST) aimed to compare the effects of the different experimental sessions (TS45, TS60 and TS75). Tests consisted of a counter movement jump (CMJ), 20 m sprint, Illinois agility test (IAT) and Hooper questionnaire. In addition, a global positioning system (GPS) was used to monitor players’ external load during the experimental sessions TS45, TS60 and TS75.

### 2.3. Testing Procedure

On the morning of the testing days MD-4 and MD, participants were asked to complete the Hooper Questionnaire. This aimed to obtain a subjective insight into individuals’ fitness level [21]. It was completed on their respective cellphone to limit teammates’ influence. They provided their subjective feeling of sleep quality the previous night, as well as ratings of fatigue, stress and delayed onset muscle soreness (DOMS). Each response was rated on a seven-point Likert scale, with responses ranging from “very, very good = 1” to “very, very bad = 7” for sleep and from “very, very low = 1” to “very, very high = 7” for fatigue, stress and DOMS. The Hooper Index (HI) was the summation of the four ratings [21].

On MD-4 and MD, just after a 15-min standardized FIFA 11+ warm-up [22], players performed in a constant order physical tests with a counter movement jump (CMJ), 20 m sprint and Illinois agility test (IAT). During the CMJ, players had to jump as high as possible, beginning in a standing position, then flexing the knees until 90° and extending the knees to jump in a continuous movement [23]. They were asked to keep their arms on their hips from standing until landing. Performance was measured using a photocell jump system (Optojump, Microgate, Bolzano, Italy) sampling at 1000 Hz, with jump height (cm) subsequently calculated by proprietary software (Optojump, Version 1.3.20.0, Microgate, Bolzano, Italy). During the 20 m sprint, 10 m and 20 m sprint times were measured using three pairs of photoelectric timing gates (Witty system, Microgate, Bolzano, Italy). Players started from a standing position as close to the timing gates as possible without triggering the cells. The IAT [24] was performed after the 20 m sprint test to assess soccer-specific speed abilities. The IAT time was measured using timing gates (Witty system, Microgate, Bolzano, Italy). Players had two trials for each test, with trials interspersed by 2 min of passive recovery. The different tests were also separated by 2 min passive recovery. The best values were used for analysis [25].

Training workload was evaluated during each experimental training session by using a GPS. Players wore a 10-Hz Fieldwizz [26] GPS alongside a 100-Hz triaxial accelerometer microsensor on every field-based training session and match. The device was located between the scapulae using a special vest. As recommended by the manufacturer, all devices were activated 15 min before data collection. GPS were distributed to the players 10 min before training session onset. Players always wore the same sensor to avoid interunit variability. GPS were turned off as soon as the training session or match was stopped. Data were downloaded and analyzed immediately after each match and training session. The manufacturer software package was used (Fieldwizz, ASI, Lausanne, Switzerland). The external load was monitored using the total distance, distance at specific speeds, accelerations and decelerations. Total distance (TD) was the distance covered by the players and was expressed in meters (m). Similarly, distances at specific speeds were expressed in m and included low-speed distance (LSD; [0–15] km·h^−1^), moderate-speed distance (MSD; [15–20] km·h^−1^), high-speed distance (HSD; [20–25] km·h^−1^) and sprint distance (SPR; >25 km·h^−1^). The total number of accelerations (ACC; >3 m·s^−2^) and decelerations (DEC; <−3 m·s^−2^) was determined.

Finally, after these experimental training sessions, players gave their rate of perceived exertion (RPE) using a Borg CR-10 scale [27]. RPE was given on individuals’ cellphones between 15 to 30 min after the end of the session [27]. 

### 2.4. Statistical Analyses

Statistical analyses were conducted using JASP (version 0.14, JASP Team 2020, University of Amsterdam, available free at https://jasp-stats.org/download/ (accessed on 9 September 2021)). Sphericity was examined by conducting Mauchly’s test. Several one-way ANOVA with repeated measurements were conducted. Parametric tests with Bonferroni’s post hoc were conducted in the case of significant training session effects for physical tests to identify any localized effect. Mean values are presented with the effect size (Cohen’s d). Effect sizes were defined as trivial (<0.2), small (0.2–0.5), moderate (0.5–0.8) and large (>0.8) [28]. Non-parametric tests (Friedman’s ANOVA and Conover’s post hoc tests) were conducted for the Hooper questionnaire and RPE. For each ANOVA, the effect sizes were calculated using the partial eta-squared ($\eta_{p}^{2}$) with values being considered as small (<0.06), moderate (0.06–0.15), or large (>0.15) [28] for parametric tests and Kendall’s W with values being considered as poor agreement (<0.20), fair agreement (0.21–0.40), moderate agreement (0.41–0.60), good agreement (0.61–0.80) and very good agreement (0.81–1.00) [29] for non-parametric tests. The statistical significance was set at *p* < 0.05. The first ANOVA was conducted for MD-4 CONTROL values (Hooper questionnaire and physical tests) to ensure a similar fitness level between conditions. The second was conducted on experimental sessions for training workloads and RPE and on MD TEST values (Hooper questionnaire and physical tests) to determine differences between the experimental training sessions.

## 3. Results

Out of 20 participants, 11 players were considered for analyses due to missed training sessions. Results of the ANOVAs are presented in Table 1. CONTROL values did not show any significant effect revealing a similar fitness level between the three experimental weeks.

### 3.1. Physical Tests

During TEST, significant differences were highlighted for CMJ, 10 m and IAT (Table 1). CMJ height was significantly greater in TS45 than in TS60 (*p* = 0.032; d = 0.946, large) and TS75 (*p* = 0.037; d = 0.830, large), while no difference existed between TS60 and TS75 (*p* = 0.851; d = 0.071, trivial) (Table 2). 10 m time showed significantly lower values in TS45 compared to TS75 (*p* = 0.033; d = 0.846, large). However, no differences were highlighted between TS45 and TS60 (*p* = 0.686; d = 0.374, small) and between TS60 and TS75 (*p* = 0.402; d = 0.471, small). 20 m time did not demonstrate significant difference between TS45 and TS60 (*p* = 0.066; d = 0.748, moderate). TS45 (*p* < 0.001; d = 1.680, large) and TS60 (*p* = 0.017; d = 0.932, large) 20 m time were both significantly lower than TS75. Additionally, IAT time showed significantly lower values for TS45 compared to both TS60 (*p* < 0.001; d = 1.583, large) and TS75 (*p* = 0.006; d = 1.080, large), while TS75 and TS60 were not different (*p* = 0.332; d = 0.503, moderate).

### 3.2. Hooper Questionnaire

During TEST, no significant difference was obtained for sleep, stress and DOMS. In contrast, significant differences were obtained for fatigue and HI (Table 1). Fatigue was significantly lower for TS45 compared to TS60 (*p* = 0.007) (Table 3). No differences were shown between TS45 and TS75 (*p* = 0.442) and between TS60 and TS75 (*p* = 0.442). HI for TEST was also significantly lower for TS45 compared to both TS60 (*p* = 0.016) and TS75 (*p* = 0.016). No difference existed between TS60 and TS75 (*p* = 1.000).

### 3.3. Field Session Performances

Statistical analyses revealed significant differences for TD, LSD, MSD, HSD, ACC, DEC (Table 1). No difference was observed for RPE. TD was significantly greater for TS75 than TS60 (*p* < 0.001; d = 1.597, large) and TS45 (*p* < 0.001; d = 3.032, large) (Table 4). TS60 was also significantly greater than TS45 (*p* < 0.001; d = 1.435, large). Same results are shown for LSD with TS75 being significantly greater than both TS60 (*p* < 0.001; d = 1.573, large) and TS45 (*p* < 0.001; d = 3.165, large). TS60 was significantly greater than TS45 (*p* < 0.001; d = 1.591, large). For MSD, TS75 was significantly greater than TS60 (*p* = 0.016; d = 0.940, large) and TS45 (*p* = 0.004, d = 1.140, large). TS60 and TS45 did not show difference (*p* = 1.000; d = 0.201, small). For HSD, a significant difference was found between TS60 and TS75 (*p* = 0.031; d = 0.850, large). TS45 and TS75 (*p* = 0.087; d = 0.709, moderate) and TS45 and TS60 (*p* = 1.000; d = 0.144, trivial) were not significantly different. For ACC, significant differences were observed between TS75 and both TS60 (*p* < 0.001; d = 1.540, large) and TS45 (*p* < 0.001; d = 1.846, large). No difference was highlighted between TS60 and TS45 (*p* = 1.000; d = 0.306, small). Concerning DEC, significant differences are shown for TS75 compared to TS60 (*p* < 0.001; d = 1.418, large) and to TS45 (*p* < 0.001; d = 1.990, large), whilst no difference existed between TS45 and TS60 (*p* = 0.114; d = 0.572, moderate).

## 4. Discussion

This study aimed to investigate the effects of different training durations during the typical MD-1 training session across French academies soccer players. In line with our hypothesis, the results highlighted that a shorter training session duration led to an increased fitness level on MD. Indeed, our results suggested that TS45 led to markedly better performance compared to the other training durations. Players’ 10 m and 20 m sprint times demonstrated that extending the duration up to 75 min led to further underperformance compared to TS45. However, as attested by CMJ and IAT, TS45 proved superior to extended sessions without distinction between TS60 and TS75.

The results of the present study could be explained by the type of exercises implemented during these training sessions. Reactivity work, also called agility, requires rapid force development, high power output and the ability to efficiently utilize the stretch-shortening cycle to accelerate and decelerate [30]. It was previously shown that increasing the weekly ACC and DEC demands led to a decreased readiness to play [31]. Furthermore, as ACC and DEC are very demanding actions, previous studies have shown that they can lead to increased muscle damage and plasma creatine kinase [15]. Indeed, on one hand, ACC were shown to require greater metabolic demand and neural activation of the working muscles compared to constant running [32]. On the other hand, DEC were shown to predominantly consist of eccentric actions leading to muscle fatigue [14], intrinsic risk factors for injury [33]. Still, ACC and DEC were similar between TS45 and TS60. This result could explain the lack of difference between these two training durations for 10 and 20 m sprints. Indeed, we might assume that a certain amount of ACC and DEC can be performed before producing a performance decrement. 

The training sessions tested here also implemented SSG. At first, SSG were introduced to replicate competitive physiological demands and technical requirements. These training exercises are generally considered very demanding [34]. It was shown that the recovery kinetic following SSG was not complete after 24 h of recovery for neuromuscular, biochemical, endocrine and mood responses [35]. Furthermore, a recent study demonstrated that fatigue following SSG could last up to 72 h during isokinetic strength tests [36]. Then, since this type of exercise imposes athletes to play under great technical pressure and fatigue, it obviously leads to significant physiological and mental fatigue [37], as confirmed by Hooper questionnaire results. Indeed, players reported increased fatigue with increasing training durations and the Hooper index was lower for TS45 (i.e., increased fitness level) than both TS60 and TS75. 

Additionally and as expected, the external load increased with duration. However, not all variables were impacted similarly by training durations. Indeed, TD and low-velocity thresholds (LSD, MSD) were more sensitive to detect external load variations than high-intensity thresholds (HSD, SPR). This result can be attributed to the implementation of this specific SSG. Indeed, as a 30 × 40 m wide SSG was implemented, we could hypothesize that high-intensity action thresholds were not easily reached. It was shown that SSG elicit predominantly low- to medium-intensity movements, while high-intensity actions are reduced in comparison to official matches [38]. Furthermore, ACC and DEC failed to differentiate TS45 and TS60 whereas these actions are crucial during reactivity drills and SSG. During these training exercises, players are expected to accelerate and decelerate over short distances. Therefore, players might not have sufficient distance to reach these high ACC and DEC thresholds and would need greater field sizes. This explanation also coincides with previous studies demonstrating that lowering ACC and DEC thresholds would better differentiate training load during SSG [39]. These results therefore suggest that low-velocity (LSD, MSD) and lower ACC/DEC (<3 m·s^−2^/>−3 m·s^−2^) thresholds should be as important as high-intensity actions during the monitoring process.

While we highlighted that this typical session was inadequate with the tapering aim, it could still match another practitioner’s aim. Indeed, a previous survey amongst French academy strength and conditioning coaches demonstrated that development was emphasized over match results [11]. Focusing on physical development requires greater training loads than a competitive result-oriented goal. Such increased training volume is usually observed during preseason (when practitioners look for physical development rather than match results) [40]. Therefore, the actual MD-1 content might be suitable when academy policies are based on long-term career development. However, this session is more likely to be tradition-based than evidence-based. The present training session seems to be part of a belief based on some potential (but erroneous) benefits. 

As competition is a top priority for elite-level teams, enabling players to be ready for competitive demands is of paramount importance. Match requires more and more physical, as long as technical and tactical demands [41]. Therefore, lowering the training load by decreasing duration or implementing more tactical sessions [42] containing drills such as set pieces [2] seems more suitable. Furthermore, because of the competitive nature of SSG, it was shown that game-based exercises could lead to increased contact injury risk [34]. This could lead to players’ absence on MD, which could lower the team’s overall level and decrease the chance of winning [43]. Additionally, implementing such demanding exercises on MD-1 could increase players’ fatigue on MD, particularly during the second half. This could expand the already high hamstring eccentric fatigue, increasing the injury risk [44]. Therefore, our study suggests that a shorter training duration is mandatory to reduce players’ fatigue and optimize performance on MD. 

Training periodization is a complex process depending on numerous factors such as players’ level, inter-individuals’ history, responses to training stimulus and the period during the pre- and in-season. Additionally, it would be of interest to explore the physiological responses associated with different exercise durations in a future study. Furthermore, our study did not explore the impact of MD-1 training duration on match performance. Due to the high variability of match performance (e.g., high-intensity performance), comparing matches can be challenging. Accordingly, the present experimental design should be replicated with a larger sample size and different contexts of training to explore the physiological responses of such training sessions alongside the impact on match performance. For instance, professional teams and different exercises or durations should be considered. It could also be of interest to consider a congested match fixture regularly present during the in-season.

## 5. Conclusions

Our study highlighted that the duration of MD-1 training session implementing reactivity and SSG impacts the physical and psychological characteristics of MD. Decreasing the duration to 45 min led to better performance on MD compared to both 60 and 75 min. In addition, practitioners should be aware that when prescribing drills such as reactivity drills and SSG, fatigue will appear from the high demands of ACC and DEC, possibly leading to underperformance and increased risks of injury more particularly on hamstring muscles during a subsequent match. Therefore, we encourage practitioners to implement lighter load exercises such as set pieces and tactical exercises on MD-1 if tapering is sought.

## Figures and Tables

**Figure 1 ijerph-19-13828-f001:**
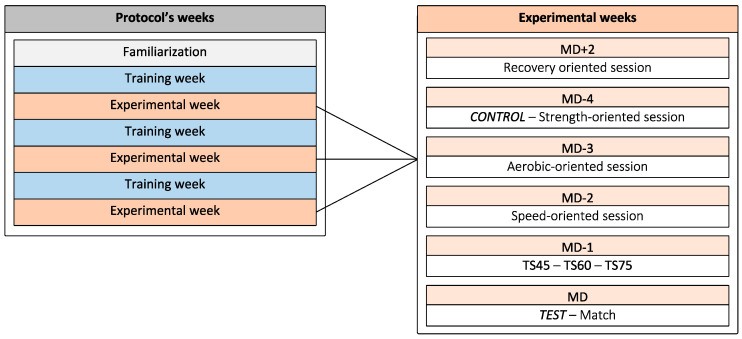
Study flowchart. Training weeks (identical contents) were performed to standardize the fitness level during the following experimental weeks. MD: Matchday; TS45: 45 min training session; TS60: 60 min training session; TS75: 75 min training session.

**Table 1 ijerph-19-13828-t001:** Results for the one-way ANOVAs.

	CONTROL	TEST
CMJ	*p*$=0.266;\;\eta_{p}^{2}$ = 0.124, moderate	*p*$=0.022;\;\eta_{p}^{2}$ = 0.317, large
10 m	*p*$=0.337;\;\eta_{p}^{2}$ = 0.103, moderate	*p*$=0.036;\;\eta_{p}^{2}$ = 0.283, large
20 m	*p*$=0.368;\;\eta_{p}^{2}$ = 0.095, moderate	*p*$<\;0.001;\;\eta_{p}^{2}$ = 0.609, large
IAT	*p*$=0.125;\;\eta_{p}^{2}$ = 0.103, moderate	*p*$<\;0.001;\;\eta_{p}^{2}$ = 0.590, large
Fatigue	*p* = 0.767; W= 0.024, poor	*p* = 0.016; W = 0.379, fair
Sleep	*p* = 0.159; W = 0.167, poor	*p* = 0.409; W = 0.081, poor
Stress	*p* = 0.717; W = 0.030, poor	*p* = 0.368; W = 0.091, poor
DOMS	*p* = 0.497; W = 0.064, poor	*p* = 0.057; W = 0.249, fair
HI	*p* = 0.629; W = 0.042, poor	*p* = 0.010; W = 0.416, moderate
	**GPS INDICATORS**	
TD	*p*$<\;0.001;\;\eta_{p}^{2}$ = 0.835, large	
LSD	*p*$<\;0.001;\;\eta_{p}^{2}$ = 0.846, large	
MSD	*p*$=0.003;\;\eta_{p}^{2}$ = 0.449, large	
HSD	*p*$=0.023;\;\eta_{p}^{2}$ = 0.314, large	
SPR	*p*$=0.120;\;\eta_{p}^{2}$ = 0.191, large	
ACC	*p*$<\;0.001;\;\eta_{p}^{2}$ = 0.683, large	
DEC	*p*$<\;0.001;\;\eta_{p}^{2}$ = 0.698, large	
RPE	*p* = 0.779; W = 0.023, poor	

The statistical significance was set at *p* < 0.05. Partial eta-squared ($\eta_{p}^{2}$) were considered as small (<0.06), moderate (0.06–0.15), or large (>0.15). CMJ: counter movement jump; IAT: Illinois agility test; TD: Total distance (m); LSD: low-speed distance ([0–15] km·h^−1^) (m); MSD: moderate speed distance ([15–20] km·h^−1^) (m); HSD: High speed distance ([20–25] km·h^−1^) (m); SPR: Sprint distance (>25 km·h^−1^) (m); ACC: accelerations (>3 m·s^−2^) (n); DEC: decelerations (<−3 m·s^−2^) (n); RPE: rate of perceived exertion.

**Table 2 ijerph-19-13828-t002:** Values of physical tests for CONTROL and TEST during the three weeks.

	CONTROL			TEST		
Indicator	TS45	TS60	TS75	TS45	TS60	TS75
CMJ (cm)	41.3 ± 5.2	43.1 ± 5.40	41.9 ± 4.3	42.7 ± 5.1 $#	40.7 ± 5.5	40.9 ± 5.7
10 m (s)	1.85 ± 0.08	1.88 ± 0.06	1.86 ± 0.08	1.83 ± 0.07 #	1.86 ± 0.07	1.90 ± 0.06
20 m (s)	3.10 ± 0.11	3.09 ± 0.08	3.13 ± 0.12	3.07 ± 0.10 #	3.13 ± 0.10 #	3.20 ± 0.10
IAT (s)	15.05 ± 0.34	15.40 ± 0.59	15.22 ± 0.44	14.82 ± 0.49 $#	15.43 ± 0.29	15.24 ± 0.33

Values are means ± SD. $: significantly greater performance than TS60 for *p* < 0.05; #: significantly greater performance than TS75 for *p* < 0.05. CMJ: counter movement jump; IAT: Illinois agility test.

**Table 3 ijerph-19-13828-t003:** Values of Hooper questionnaire for CONTROL and TEST during the three weeks.

	CONTROL			TEST		
Indicator	TS45	TS60	TS75	TS45	TS60	TS75
Fatigue	2.82 ± 1.17	2.55 ± 0.82	2.55 ± 1.21	2.27 ± 0.47 $	3.36 ± 1.12	2.73 ± 0.90
Sleep	2.73 ± 1.56	2.91 ± 1.30	1.91 ± 1.04	1.73 ± 1.27	2.09 ± 0.70	1.91 ± 0.70
Stress	1.27 ± 0.65	1.36 ± 0.50	1.27 ± 0.47	1.09 ± 0.30	1.45 ± 0.93	1.27 ± 0.47
DOMS	2.27 ± 1.35	1.91 ± 1.22	2.36 ± 1.36	2.55 ± 0.82	4.09 ± 1.92	3.82 ± 1.25
HI	9.09 ± 2.47	8.73 ± 2.80	8.09 ± 3.59	7.64 ± 1.50 $#	11.00 ± 3.49	9.73 ± 2.41

Values are means ± SD (A.U.). $: significantly lower value than TS60 for *p* < 0.05; #: significantly lower value than TS75 for *p* < 0.05. DOMS: Delayed Onset Muscle Soreness; HI: Hooper Index.

**Table 4 ijerph-19-13828-t004:** Values of GPS indicators and RPE for MD-1 during the three weeks.

Indicator	TS60	TS75	TS45
TD	3842.4 ± 225.8 @	4720.2 ± 343.4 $@	3053.9 ± 553.6
LSD	3594.0 ± 188.3 @	4358.4 ± 297.6 $@	2820.8 ± 485.8
MSD	215.2 ± 66.2	304.6 ± 79.0 $@	196.1 ± 100.4
HSD	31.7 ± 23.5	55.4 ± 21.5 $	35.7 ± 25.7
SPR	2.4 ± 5.0	3.9 ± 5.6	0 ± 0
ACC	48.3 ± 10.3	73.9 ± 17.8 $@	43.2 ± 10.5
DEC	39.5 ± 12.1	54.6 ± 11.1 $@	33.4 ± 7.0
RPE	5.73 ± 1.27	5.91 ± 0.90	5.82 ± 0.60

Values are means ± SD. @: significantly greater performance than TS45 for *p* < 0.05; $: significantly greater performance than TS60 for *p* < 0.05. TD: Total distance (m); LSD: low-speed distance ([0–15] km·h^−1^) (m); MSD: moderate speed distance ([15–20] km·h−1) (m); HSD: High speed distance ([20–25] km·h−1) (m); SPR: Sprint distance (>25 km·h^−1^) (m); ACC: accelerations (>3 m·s^−2^) (n); DEC: decelerations (<−3 m·s^−2^) (n); RPE: rate of perceived exertion.

## Data Availability

The data presented in this study are available on request from the corresponding author.

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
