# Peer review of "Physical Impact of a Typical Training Session with Different Volumes on the Day Preceding a Match in Academy Soccer Players"

_ijerph, 2022, doi:10.3390/ijerph192113828_

Round 1

Reviewer 1 Report

This is a very interesting study with immediate practical applications on youth soccer (and consequently on soccer in general).

The main finding was that the shorter training session (TS45) on MD-1 results in the best performance (20 m sprint, CMJ, Illinois test) on MD, while increasing session duration to 60 or 75 min impairs performance on MD.

 Please address the following comments:

Line 15: This sentence is not clear. It is not evident that "Durations decreased and increased by 25%". Please clarify what you mean. I guess that you are saying that compared to 60 min, a duration of 45 min is 25% lower, while a duration of 75 min is 25% higher. Please state this clearly.

Line 94: What was the structure of the SSG when the duration was 18 min? How many repetitions of 2min work:2 min rest did the players perform? This is slightly confusing.

Lines 95-96: How did you perform the "blinding" of the players? Did you explain to them that you would use different durations in the three conditions or not?

Did you ensure that the other day's training load (MD-4, -3, -2) were similar in the three experimental weeks? Please provide a brief description of the training load and content.

The Hooper Questionnaire contains questions regarding muscle soreness and fatigue. Please explain which elements of the Hooper index were lower in the TS45 condition. Could this be an effect of the training during the previous days? (MD-4 etc.).

Lines 164-167: please justify the use of these speed limits for the different activities? The are slightly different from the "standard" values.

Table 4: Please do not use two decimal points for distances and accelerations. I would suggest to either use one or no decimal points for all distances.

Table 4: Sprint distance is 2.44, 3.93 or 0 meters. Please justify these very low values.

Match Day: I am not sure if the players actually played a match on MD. From what I understand, the study was conducted during the winter season (before the break), and therefore there was a match every week. I have some questions to clarify some important details here:

a) did the players play a game each week, irrespective of the Experiment (I am looking at Figure 1).

b) was the MD the same day of the week in all conditions? If not, how did you handle "shorter" or "longer" weeks?

c) Have you recorded GPS performance on MD? If yes, did any of the interventions have an effect on GPS running performance?

d) Table 1: It is not clear which condition is different from another in the "TEST" column. Please clarify

Reviewer 2 Report

A review of the paper submitted to the Int Journal of Environmental Research and Public Health entitled “Physical impact of a typical training session with different volumes on the day preceding a match in academy soccer players” In the past decade or two we have seen a lot of research regarding postactivation performance enhancement (PAPE), a phenomenon in which an acute bout of high intensity exercise is followed by an improvement in strength, speed and power performance. This acute performance enhancement has been observed within minutes of activation (2-12min). PAPE strategies are effective in improving explosive motor performance, but because of the small window of opportunity they are difficult to include in competition. Considering the above scientists and coaches have been exploring another window of potentiation lasting for up to 48h following a pre-competition training session. Delayed potentiation (priming) has been observed in numerous neuromuscular performance measures and thus could be employed in pre-competition strategies to optimize performance. This has been evaluated in simple motor tasks such as jumping, throwing and sprinting, however the idea has not been tested extensively in team sport games, where multiple motor abilities and skills determine performance during competition. Thus evaluating the impact of different volumes of a pre-competition training session on game performance seems fully justified. Manipulating different workloads during the training session preceding competition and evaluating key neuromuscular performance measures 24h later may help coaches optimize periodization to reach peak performance during game time. Thus a proper periodization of training loads seems to be the key factor determining performance during a competitive season in team sports such as soccer. The Introduction is quite short but the objectives of the research are clearly presented and a brief literature review is performed to highlight the significance of the topic. A hypothesis has also been presented. The methods section is written very well and consists of participants’ description, the study design, testing procedures and statistical tools used in the analysis of results. The number of participants was rather low (20) yet it is fully justified as an elite group of youth players was tested on 4 different occasions, with the objective to compare the physical and physiological strain on the body depending on the workload of the last training session before MD. The study flowchart introduces the reader into the structure of the research protocol and the weekly microcycle preceding MD. The training loads were similar in consecutive weeks of the experiment thus fitness level of the players was roughly the same when attending the pre-game training unit. The tests evaluating neuromuscular performance were well chosen and included power (CMJ), speed (20m sprint) and agility (IAT). The Hooper’s Questionnaire was applied to determine the level of fatigue, stress and muscle soreness. This is a subjective form of evaluating these sensations thus an objective physiological marker such as HR or HRV or a biochemical one informing about muscle damage such as CK or LDH activity could have been used, yet the amount of variables was rather high thus this was not absolutely necessary. The training workload was measured quite precisely through GPS units, what allowed for a comparison between the 3 different exercise protocols. Besides the objective evaluation of workloads by total distance covered as well as distance covered in slow, moderate, high sped running, sprinting and accelerations and decelerations were recorded during the training sessions that included SSG with different volume of work. The results are presented in 4 tables, including the significance level. The discussion is a strong part of the manuscript, where the authors present the main findings of the research and confront them with recent 40 references. The main finding of the research is highlighted in the first paragraph. It states that shorter training sessions on the day preceding competition reduce players fatigue and increases delayed potentiation to optimize performance on MD. The authors could add a few lies on the limitations of the study. The conclusions are both sound scientifically and valuable practical clues for coaches and athletes.

Reviewer 3 Report

Dear authors,

This research article is very interesting and has high practical implications in the field of training science. However, several limitations should be addressed. I suggest that the statistical analysis should be re-conducted regarding measured ordinal scales. The section of discussion seems to be speculative because of the absence of measured physiological indicators. I suggest that the authors should satisfactorily address all my comments and several limitations before this article will be published in IJRPH.

Abstract

The abstract should be changed according to the main text (you can see comments for the main text).

Introduction

Line 42: In a recent survey in French academics…. Is this for all sports disciplines or only for soccer players?

Line 47: ……lead to performance decrements (neuromuscular fatigue [14]). The authors should shortly explain the mechanism of neuromuscular fatigue. Furthermore, please add information about different SSGs’ intensities in detail (e. g. physiological demands) according to previous studies.

Line 53-54: References are missing or if there is no reference for that, remove these sentences.

Line 56: In which athletes/players?

Materials and Methods

Line 65-66: Please add body fat data in % and why/how twenty players (n = 20) were recruited for this study.

Line 68-69: please add the main play positions of twenty players.

Line 70: …..food and water intake -> please change to “nutritional status”

Line 72: which local ethic committee is this? Please add this information.

Line 79: volunteers -> please change this to “participants” in the entire manuscript.

Line 89-90: Why was this SSG method selected?

Line 150-152: References are missing.

Line 174-187: why were Hooper questionnaire and RPE items conducted by the parametric test to compare? These are ordinal scales right?! I would like to suggest that ordinal scales (rank) should be performed by a non-parametric test. Please explain this selection of the statistical analysis.

Results

I would like to recommend a graphical representation with each value as a statistical analysis of the physical test e. g. Table 2. Furthermore, ordinal scales should be performed by non-parametrical tests and calculate the appropriate effect sizes.

Discussion

Line 254: Also, add “soccer players”

Line 260-261: These sentences should be involved at the end of the discussion.

Line 267-270: However, these physiological parameters were not directly measured. Therefore, I suggest that it will be possible to evaluate the physiological responses associated with different exercise durations in a further study. Please consider this limitation in this paragraph.

Line 286-287: Here, please reconsider your statistical analyses. Also, physiological aspects were not involved in this study.

Generally, authors interpreted factors speculative in sections of discussion and conclusion without e.g. measurements of physiological factors. Therefore, I expect that this point should critically be discussed.

Round 2

Reviewer 3 Report

Dear Authors,
All responses to reviewers' comments have been addressed satisfactorily. Thank you for the efforts of the authors and congratulation.